# Slot-guided Volumetric Object Radiance Fields

**Di Qi**
MEGVII Technology Inc.
qidi@megvii.com

**Tong Yang**
MEGVII Technology Inc.
yangtong@megvii.com

**Xiangyu Zhang**[*]
MEGVII Technology Inc.
zhangxiangyu@megvii.com

## Abstract

We present a novel framework for 3D object-centric representation learning. Our approach effectively decomposes complex scenes into individual objects from a single image in an unsupervised fashion. This method, called slot-guided Volumetric Object Radiance Fields (sVORF), composes volumetric object radiance fields with object slots as a guidance to implement unsupervised 3D scene decomposition. Specifically, sVORF obtains object slots from a single image via a transformer module, maps these slots to volumetric object radiance fields with a hypernetwork and composes object radiance fields with the guidance of object slots at a 3D location. Moreover, sVORF significantly reduces memory requirement due to small-sized pixel rendering during training. We demonstrate the effectiveness of our approach by showing top results in scene decomposition and generation tasks of complex synthetic datasets (e.g., Room-Diverse). Furthermore, we also confirm the potential of sVORF to segment objects in real-world scenes (e.g., the LLFF dataset). We hope our approach can provide preliminary understanding of the physical world and help ease future research in 3D object-centric representation learning.

## 1 Introduction

As humans, we can understand scenes, perceive discrete objects within it, and interact with these objects in a 3D environment. This object-centric, geometric understanding of the 3D world is a fundamental ability in human vision [1]. In computer vision, researchers attempt to replicate this fundamental ability in machine learning models with a high interest, due to its wide application ranging from robotics [2] to autonomous navigation. To this end, machine learning models should bear two characteristics: unsupervised representation learning manner and 3D-aware generative mode [3].

Recently, with the advances of neural radiance fields (NeRFs) [4] and representation learning [5, 6, 7], there are some works [8, 9, 3] to achieve these two characteristics. These works learn to decompose objects and understand 3D scene geometry from RGB supervision via novel view synthesis in an unsupervised manner. To this end, volumetric-based methods [8] utilize volume rendering mechanism to implement a 3D-aware differentiable generative process without supervision. To avoid high computational cost of volumetric-based methods, light-field based methods [9, 3] use light field formulation [10] to scale its application to large numbers of objects in a scene. However, existing works suffer from some limitations. Light-field based methods lack strict multi-view consistency [10] and fall into mask bleeding issues [3], which are harmful for object-centric representation learning. Besides, volumetric-based methods fail to decompose scenes caused by attention rank collapse [8] and do not perform well in complex multi-object scenes.

In this paper, we propose a novel framework for 3D object-centric representation learning, alleviating the issues of existing works. Our method, called s̲lot-guided V̲olumetric O̲bject R̲adiance

---

[*]Corresponding author.

37th Conference on Neural Information Processing Systems (NeurIPS 2023).

Fields (sVORF), adopts volumetric rendering to synthesis novel views and use object slots as a guidance to compose volumetric object radiance fields. Concretely, we firstly use an efficient transformer module to extract object slots from a single image and learn object-aware slot features with the help of self-attention mechanisms [11]. Then we utilize a hypernetwork to map these slots to volumetric object radiance fields. Finally, at each 3D location, we compose object radiance fields with the guidance of object slots for volumetric rendering. Thus, sVORF can avoid instinctive limits of light field formulation and resolve mask bleeding issues. Meanwhile, with the benefits from the guidance of object slots, sVORF can learn 3D-aware slot features and facilitate network optimization, alleviating the issues faced by existing volumetric-based methods. Moreover, instead of rendering whole image [8, 9], sVORF only render a small amount of image pixels during the training phase, which significantly reduces the demand for training resources.

To validate the effectiveness of sVORF, we conduct experiments on four synthetic datasets to assess the ability of scene decomposition (e.g., segmentation in 3D) and scene generation(e.g., novel view sythesis, scene editing in 3D). Our results demonstrate that sVORF can precisely decompose 3D scenes into individual objects and produce high-quality novel view images. Specially, sVORF outperforms other state-of-the-art methods by a significant margin in complex multi-object scenes. In ablation experiments, we also analyze the effects of core components and show their strength. Besides, we show the robustness of sVORF on unseen object appearance and unfamiliar spatial arrangements. Furthermore, we extend our validation process to complex real-world LLFF data, confirming that our approach can segment objects in complex scenes with high accuracy.

To sum up, our contributions are three fold. First, we introduce a novel approach for 3D-centric representation learning, named sVORF, that effectively decompose objects from a single image. Second, our slot-guided scene composition method avoids the shortcomings of existing methods and significantly reduces the memory requirements during training phase. Third, we validate the effectiveness of our proposed method on synthetic datasets and confirm the extendability of our approach on real-world scenes.

## 2 Related Work

### 2.1 Neural Scene Representation and Rendering

Neural scene representations parameterize 3D scenes with a deep network that map *xyz* coordinates to signed distance functions or occupancy fields. Equipped with differentiable rendering functions, they can be optimized using only 2D images, relaxing the requirement of 3D ground truth. In particular, Neural Radiance Fields (NeRFs) [4] can use an MLP to compute radiance values (color and density) for a given 5D coordinate (spatial location $(x, y, z)$ and viewing direction $(\theta, \phi)$) and produce novel views with remarkably fine details. Numerous subsequent works have been introduced to address some its shortcomings and expand its applications, including rendering acceleration [12, 13, 14, 15, 16, 17, 18, 19, 20], NeRF with few images [21, 22, 23, 24], 3D reconstruction [25, 26, 27, 28, 29] and 3D scene semantic understanding [30, 31, 32, 33, 34]. However, these volumetric methods need hundreds of evaluations for a ray, leading to an expensive cost for rendering. To address this issue, Light Field Networks (LFN) [10, 24] directly map an input ray to an output color, making only a single evaluation of the MLP per ray.

### 2.2 3D Object-centric Representation Learning

Driven by the effectiveness of NeRF, recent research has attempted to combine 2D self-supervised object-centric models [5, 6, 7] with neural scene representations to decompose a 3D scene to individual objects. Earlier work [35] utilizes a slot-based encoder and NeRFs as 3D representations to decompose 3D scene with extra multi-view dense depth supervision. Likewise, uORF [8] explicitly model the separation of objects and background with only images in training to address complex scenes. In order to avoid expensive cost of volume rendering, OSRT [3] and COLF [9] further replace the volumetric parametrization with a light field formulation. However, existing methods face some limitations. ObSuRF [35] need dense depth as supervision and can not handle complex scenes, while uORF [8] encounters expensive computation cost during training. Although light field formulation method [3, 9] resolve computation cost issues, they lack strict multi-view consistency [10] and easily fall into mask bleeding issues [3].

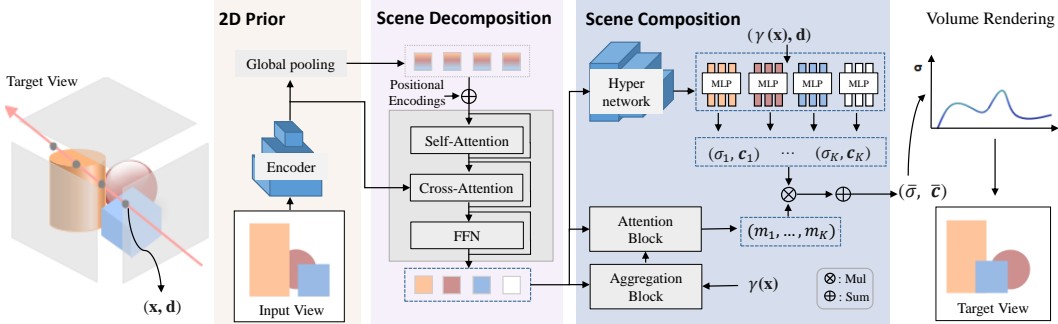

Figure 1: **sVORF overview**. The image encoder $E_\theta(\mathbf{I})$ processes the source view of a scene to generate 2D image features that serve as a prior. Next, these features are fed into the *Scene Decomposition* module to infer object and background slots. A hypernetwork then maps these slots to volumetric object radiance fields. Finally, the object slots provide guidance for the recombination of object radiance fields to render arbitrary views with 3D-consistent object decomposition.

Different from the above methods, our method adopts volume representation to parameterize 3D scenes, avoiding the limits of light field formulation. Also, our approach has low computation cost by only sampling a small amount of rays and avoids using depth supervision during training. Moreover, our method can be applied to object segmentation in real-world scenes (e.g. LLFF dataset) [36].

## 3 Method

The aim of our method is to decompose a scene to a set of object-centric 3D representations given a single input image. An overview of our approach is provided in Figure 1. We firstly extract image features from an input image and decompose object slots from image features. Then we map these slots to volumetric object radiance fields and compose these object radiance fields with the guidance of object slots for novel view synthesis.

### 3.1 Preliminaries: NeRF

We begin by briefly reviewing the Neural Radiance Fields (NeRFs). A radiance field encodes a scene as a continuous volumetric radiance field $f$. The input to $f$ is a location $\mathbf{x} \in \mathbb{R}^3$ and a viewing direction $\mathbf{d} \in \mathbb{R}^2$, while the output is an RGB color value $\mathbf{c} \in \mathbb{R}^3$ and a volume density $\sigma \in \mathbb{R}^+$. NeRFs parameterize $f$ as an MLP $\Phi_\phi$. To make MLP learns high-frequency functions, the input coordinates $\mathbf{x}$ and viewing directions $\mathbf{d}$ are mapped into a higher dimensional space with sinusoids function $\gamma(\cdot)$ before being passed into the MLP [37]. This process, known as positional encoding, allows the MLP to better capture high-frequency scene content. Therefore, the function $\Phi_\phi$ can be formulated as:

$$\Phi_\phi : (\gamma(\mathbf{x}), \gamma(\mathbf{d})) \mapsto (\sigma, \mathbf{c}) \tag{1}$$

Given N sampled points along a ray $\mathbf{r}$ and its predicted color and volume density $\{(\mathbf{c_i}, \sigma_i), i \in \{1, \ldots, N\}\}$, the expected color $C(\mathbf{r})$ of camera ray $\mathbf{r}$ can be derived from volume rendering:

$$C(\mathbf{r}) = \sum_{i=1}^{N} T_i \left(1 - \exp\left(-\sigma_i \delta_i\right)\right) \mathbf{c}_i, \quad T_i = \exp\left(-\sum_{j=1}^{i-1} \sigma_j \delta_j\right) \tag{2}$$

where $\delta_i$ indicates the distance between adjacent samples. The reconstruction loss between the rendered and true pixel colors is used during training to optimize the parameters $\phi$ of MLP.

### 3.2 Scene Decomposition from 2D Prior

Following [35, 9], we define the scene as a combination of $K$ entities, where the first $K-1$ represent the objects and the last one represents the background. To obtain these entities, we first extract image feature as a 2D prior using an encoder $E(\mathbf{I})$ from a scene image $\mathbf{I}$. Instead of using slot attention module as existing methods, we adopt an efficient transformer module $T$ to infer object and

background slots from image feature, termed as $\mathbf{S} = \{\mathbf{s}_i\}_{i=1}^K$. Compared to slot attention module, this transformer module is simple and easy to train without Gated Recurrent Unit (GRU) block. For this transformer module, we take a global image feature $\bar{v} = avg\_pool(E(\mathbf{I}))$ as initial object slots $\mathbf{Z} = \{\mathbf{z}_i\}_{i=1}^K$. Combining with $K$ learned positional encodings, these slots explicitly model all pairwise interactions between all slots and learn object-aware features via self-attention, which avoids inherent ambiguities in estimating color and geometry at occluded views, see Section 4.4 for details. Then these slots bind and explain the input image representation with cross-attention to image feature. Finally, the slots are transformed to $\mathbf{S} = \{\mathbf{s}_i\}_{i=1}^K$ via feed forward network (FFN). The whole process can be formulated as follows:

$$\mathbf{S} = T(\mathbf{Z}, E(\mathbf{I})) \tag{3}$$

### 3.3 Scene Composition

Given the infered slots $\mathbf{S} = \{\mathbf{s}_i\}_{i=1}^K$, we recompose these slots to novel views of the same scene, which is critical for achieving 3D object-centric representation learning. To achieve it, there are two forms: Spatial Broadcast (SB) [35, 8, 9, 3] and Slot Mixers (SM) [3]. SB and SM do scene composition on RGB space and feature space, respectively. SB can utilize 3D geometric bias (3D point or 3D ray) in the scene composition, facilitating the network optimization. Conversely, due to composition on feature space, SM is hard to optimize without 3D geometric bias. But SM can learn 3D-aware slot features, which is useful for scene decomposition. Combining the advantages of two forms, we propose a new method for scene composition. We transform slots into volumetric neural radiance fields to utilize explicit geometric bias and avoid the limits of light field formulation [10]. Then, we compose all volumetric neural radiance fields with the guidance of slots, making slot features 3D-aware.

**Objects as Neural Radiance Fields**    To transform a slot to its radiance field, we utilize a hyper-network [38, 39] $H$ to map the slot $\mathbf{s}_i$ directly to the parameters $\phi_i$ of the associated object Neural Radiance Field. The mapping process is formulated as:

$$\phi_i = H(\mathbf{s}_i) \tag{4}$$

where $i = 1, \cdots, K$, $K$ is the number of object slots.

With the radiance field $\Phi_{\phi_i}$, we can map a location $\mathbf{x} \in \mathbb{R}^3$ with a viewing direction $\mathbf{d} \in \mathbb{R}^2$ to a tuple of color $\mathbf{c}_i$ and density $\sigma_i$ of corresponding object:

$$\Phi_{\phi_i} : (\gamma(\mathbf{x}), \gamma(\mathbf{d})) \mapsto (\sigma_i, \mathbf{c}_i) \tag{5}$$

In this way, we transform feature space to RGB space, introducing 3D geometric bias for scene composition.

**Composing Mechanism**    Given K Object NeRFs $\{\Phi_{\phi_i}\}_{i=1}^K$, we compose their outputs $\{\sigma_i\}_{i=1}^K$ and $\{\mathbf{c}_i\}_{i=1}^K$ at a 3D location with the guidance of object slots. Specifically, we firstly leverage a feature aggregation block $D^c$ to gather object slots and obtain an aggregate feature $\mathbf{z}$ for query location $\gamma(\mathbf{x})$:

$$\mathbf{z} = D^c(\gamma(\mathbf{x}), \mathbf{S}) \tag{6}$$

where $D^c$ is a cross-attention layer [40] network. Then we pass $\mathbf{z}$ to an attention block $D^a$ and compute a normalized dot-product similarity between $\mathbf{z}$ and each slot feature in $\mathbf{S}$:

$$\mathbf{m} = D^a(\mathbf{z}, \mathbf{S}) \tag{7}$$

where $\mathbf{m} = (m_1, \cdots, m_K)$, $D^a$ is a cross-attention layer without linear operation. Finally, we compute the combined density $\bar{\sigma}$ and color $\bar{\mathbf{c}}$ as follows:

$$\bar{\sigma} = \sum_{i=0}^K m_i \sigma_i, \quad \bar{\mathbf{c}} = \sum_{i=0}^K m_i \mathbf{c}_i \tag{8}$$

Note that our model can train with a small amount of sample rays, leading to a significant reduction for computation and memory cost during training. We speculate that this characteristic benefits from our composing mechanism. In order to decompose small objects from a scene, it is essential to sample much more rays to cover the regions of these small objects, resulting in heavy training cost. However, our composing mechanism can perform well on small objects with a small amount of rays, addressing this large sampling cost, see Section 4.4.

### 3.4 Loss Functions

**Reconstruction Loss** We train our model across multiple scenes and only take view images as the supervisory signal. The reconstruction loss is formulated as:

$$\mathcal{L}_{\text{recon}} = \sum_{\mathbf{r} \in \mathcal{R}} \|C(\mathbf{r}) - C_{gt}(\mathbf{r})\|^2 \tag{9}$$

where $\mathcal{R}$ is the set of rays in each batch, and $C_{gt}(\mathbf{r})$ is the groundtruth color.

**Connectivity Regularization** We observe that some object radiance fields exist semi-transparent clouds, especially when the number of slots is much larger than the total number of objects in a scene. To solve this issue, we apply a connectivity regularization $\mathcal{L}_{\text{connect}}$ to each $\Phi_{\phi_i}$ by referring to the distortion loss presented in Mip-NeRF360 [41]:

$$\mathcal{L}_{\text{connect}}(\mathbf{t}, \mathbf{w}) = \sum_{i,j} w_i w_j \left| \frac{t_i + t_{i+1}}{2} - \frac{t_j + t_{j+1}}{2} \right| + \frac{1}{3} \sum_i w_i^2 (t_{i+1} - t_i) \tag{10}$$

where $\mathbf{w} = \{T_i(1 - \exp(-\sigma_i \delta_i))\}_{i=1}^N$ is the weights along a ray and $\mathbf{t}$ is the normalized ray distance.

**Total Loss** The overall training loss function is formulated as follows:

$$\mathcal{L} = \mathcal{L}_{\text{recon}} + \lambda_{\text{connect}} \mathcal{L}_{\text{connect}} \tag{11}$$

where $\lambda_{\text{connect}}$ is the scale to balance the connectivity regularization $\mathcal{L}_{\text{connect}}$, which is set to be 0.01 in our experiments.

## 4 Experiments

**Datasets** Following uORF [8], we experiment on several datasets in increasing order of complexity.

*CLEVR-567* [8]: The CLEVR [42] dataset is a widely used benchmark for evaluating object decomposition in computer vision. CLEVR-567 is a multicamera variant of this dataset with 1,000 scenes for training and 500 scenes for testing. Each scene consists of with 5-7 CLEVR objects that randomly positioned and oriented with a clean background. The objects in the scenes are comprised of three geometric primitives: cubes, spheres, and cylinders. We follow uORF's setup in using a "Rubber" material with largely diffuse properties.

*CLEVR-3D* [35]: This dataset is also a variant of CLEVR dataset, in which each scene consists of 3-6 basic geometric shapes of 2 sizes and 8 colors. In particular, each scene includes 3 fixed views: the two target views are the default CLEVR input view rotated by $120°$ and $240°$, respectively. Following ObSuRF [35], we train 35k scenes and test on the first 320 scenes of each validation set [43, 6].

*Room-Chair* [8]: This dataset contains 1,000 scenes designated for training and 500 for testing. In this dataset, each scene includes 3-4 chairs of identical shape with 3 different textures background.

*Room-Diverse* [8]: This dataset is an upgraded Room-Chair. Each scene contains 4 distinct chairs, whose shape chosen randomly from ShapeNet [44] chair shapes, and a range of background that is selected from 50 unique textures. There are 5,000 scenes for training and 500 for testing.

*MultiShapeNet (MSN)* [35]: This dataset comprises 11,733 distinct shapes, with each scene populated by 2-4 objects with different categories sourced from the ShapeNetV2 3D model dataset.

*Local Light Field Fusion (LLFF)* [36]: This dataset includes real scene scenarios with complex forground and background, making it highly challenging. We specifically utilize the forward-facing scenes *Flower* and *Fortress* from the LLFF dataset, with each scene consisting of 27 training images and 6 testing images.

**Baselines** We compare our model with a 2D object-centric learning method Slot-Attention [43] and four competitive 3D methods, namely uORF [8], COLF [9], ObSuRF [35] and OSRT [3], in terms of scene decomposition and novel view synthesis. Both COLF and OSRT are based on light field, while uORF and ObSuRF employ volumetric parameterization, which is similar to our approach.

Table 1: Scene segmentation results. **Bold** and Underline indicate state-of-the-art (SOTA) and the second best.

| Model | CLEVR-567 | | | Room-Chair | | | Room-Diverse | | |
|---|---|---|---|---|---|---|---|---|---|
| | 3D metric | 2D metric | | 3D metric | 2D metric | | 3D metric | 2D metric | |
| | NV-ARI ↑ | ARI ↑ | Fg-ARI↑ | NV-ARI ↑ | ARI ↑ | Fg-ARI↑ | NV-ARI ↑ | ARI ↑ | Fg-ARI↑ |
| Slot Attention [43] | N/A | 3.5 | **93.2** | N/A | 38.4 | 40.2 | N/A | 17.4 | 43.8 |
| uORF [8] | **83.8** | **86.3** | 87.4 | 74.3 | 78.8 | 88.8 | 56.9 | 65.6 | 67.9 |
| COLF [9] | 46.6 | 59.5 | 92.6 | 83.5 | 83.9 | 92.4 | 54.5 | 70.7 | 71.7 |
| sVORF | 81.5 | 82.7 | 92.0 | **87.0** | **87.8** | **92.4** | **75.6** | **78.4** | **86.6** |

Table 2: Results on CLEVR-3D dataset. [†]Model is re-evaluated using the unified test set for a fair comparison. **Bold** and Underline indicate state-of-the-art (SOTA) and the second best.

(a) **3D Segmentation**.

| Model | Supervision | ARI* ↑ | Fg-ARI*↑ |
|---|---|---|---|
| ObSuRF [35] | image+depth | **94.6** | 95.7 |
| OSRT[†] [3] | image | 42.7 | **97.0** |
| sVORF | image | 86.0 | 96.3 |

(b) **Novel View Synthesis**.

| Model | LPIPS ↓ | SSIM ↑ | PSNR ↑ |
|---|---|---|---|
| ObSuRF [35] | N/A | N/A | 33.69 |
| OSRT[†] [3] | 0.0367 | 0.9719 | 36.74 |
| sVORF | **0.0258** | **0.9759** | **37.52** |

**Metrics**   To evaluate the quality of novel view synthesis, we use Learned Perceptual Image Patch Similarity (LPIPS) [45], Structural Similarity Index (SSIM) [46], and Peak Signal-to-Noise Ratio (PSNR). For scene segmentation in 3D, we measure clustering similarity using Adjusted Rand Index (ARI). The ARI score ranges from 0 to 1, with a score of 0 indicating random segmentation and a score of 1 indicating perfect segmentation. For a fair comparison, we evaluate two types of 2D ARI metrics: ARI and FG-ARI, as well as three types of 3D ARI metrics: NV-ARI, ARI*, and FG-ARI*. Specifically, both ARI and FG-ARI are computed on the source image to facilitate comparison with 2D methods. The FG-ARI is further calculated solely on the foreground regions, using ground-truth data. Similarly, ARI* and FG-ARI* are calculated on all images. Lastly, the NV-ARI is computed on synthesized novel views.

## 4.1   Scene Segmentation in 3D

**Setup**   Given the soft slot masks of 3D locations, 2D segmentation masks are inferred through volume rendering. To clarify, we begin by mapping the pixel $\mathbf{p}$ in the rendered view to a ray $\mathbf{r}$ for sampling N points along it. Then, we calculate the soft mask $\mathbf{m}$ of each sampling point $\mathbf{x}$ at all object NeRFs and derive the composite density $\bar{\sigma}$ using the mask information. Finally, we render the segmentation mask along the ray based on the composite density to yield the segmentation of pixel $\mathbf{p}$.

**Results**   We compare our method with uORF and COLF, and present the results in Table 1. The comparison reveal that our method outperforms all baselines in terms of NV-ARI and ARI values, particularly the NV-ARI, in both Room-Chair and Room-Diverse scenes. It provides evidence that sVORF can effectively identify 3D objects from a single image with better multi-view consistency. Furthermore, in the more complex Room-Diverse scene, our approach achieves comprehensive and distinct improvements over other methods, indicating the robustness of our design. Moreover, we report the performance on CLEVR-3D in Table 2a to ensure equitable comparisons with ObSuRF and OSRT. Compared with OSRT, sVORF achieves significantly higher ARI* and similar FG-ARI* without using depth information, which further proves that our volume parameterization can alleviate mask bleeding problems, as shown in Figure 3. However, we encounter exceptions in the ARI and NV-ARI scores on the CLEVR-567 dataset, and the ARI* on CLEVR-3D dataset. Our benchmark scores are slightly lower than those of uORF and ObSuRF. We attribute this to the penalty imposed on reconstructing shadows of foreground objects, which are not included in the ground truth object masks. This inadequacy is illustrated in Figure 3. Since the light source remains fixed in the CLEVR dataset, it is straightforward to model the shadow as a translucent layer belonging to the object. Nonetheless, the high FG-ARI indicates that our proposed method is proficient in forming factorized representations, which can segment objects in a scene effectively.

Table 3: Comparison on novel view synthesis from a single image.

| Model | CLEVR-567 | | | Room-Chair | | | Room-Diverse | | |
|---|---|---|---|---|---|---|---|---|---|
| | LPIPS ↓ | SSIM ↑ | PSNR ↑ | LPIPS ↓ | SSIM ↑ | PSNR ↑ | LPIPS ↓ | SSIM ↑ | PSNR ↑ |
| uORF [8] | 0.0859 | 0.8971 | 29.28 | 0.0821 | 0.8722 | 29.60 | 0.1729 | 0.7094 | 25.96 |
| COLF [9] | 0.0608 | 0.9346 | 31.81 | **0.0485** | 0.8934 | 30.93 | **0.1274** | 0.7308 | 26.02 |
| sVORF | **0.0211** | **0.9701** | **37.20** | 0.0824 | **0.8992** | **33.04** | 0.1637 | **0.7825** | **29.41** |

Furthermore, to validate our method on more challenging scenarios, we conduct experiments on the MSN dataset and show the results in Table 4. Compared with ObSuRF, sVORF achieves significantly higher Fg-ARI and comparable ARI without using depth information, which demonstrates the model's ability to decouple more complex scenarios. Qualitative results are shown in Figure 2.

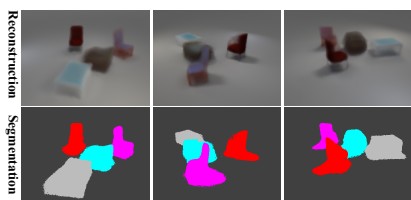

Figure 2: Qualitative results of sVORF on MSN.

Table 4: Comparison on MSN dataset.

| Model | ARI* ↑ | Fg-ARI* ↑ | PSNR ↑ |
|---|---|---|---|
| ObSuRF [35] | **64.1** | 81.4 | 27.41 |
| sVORF | 63.4 | **84.1** | **30.51** |

## 4.2 Novel View Synthesis

**Setup**   For each test scene, we reserve one view as an input and use the other views to evaluate the the quality of reconstruction.

**Results**   In Table 3, sVORF outperforms all baselines on most metrics, despite not employing the coarse-to-fine training schedule. These findings suggest that our method is effective in producing high-quality images. The slightly lower LPIPS performance on Room-Chair and Room-Diverse can be attributed to the lack of perceptual loss in the sVORF training process, which does not explicitly optimize the LPIPS performance. Moreover, there exists a trade-off between structural and perceptual loss within the model. The excellent SSIM performance of our model indicates that the sVORF can generate complex object shapes. Specifically, as illustrated in Figure 3, our method generated diverse chair shapes with higher accuracy than COLF and uORF, even though it did not fully recover the background texture. In addition, the results presented in Table 2b indicate that sVORF outperforms OSRT in all metrics, despite using nearly half of the parameters as OSRT (48.73M v.s. 83.11M). Without the addition of depth information, our method synthesizes multi-view images with higher quality than obsurf, as shown in Table 4.

## 4.3 Scene Design and Editing in 3D

**Setup**   We examine the potential of our method for basic scene editing on the Room-Chair dataset. Following uORF, our investigation involves two types of modifications, namely, moving objects and changing backgrounds. For editing the position of a foreground object, we move all query point coordinates on its object NeRF, based on the targeted movement. In order to relocate the slot, an affine transformation is applied to the 3D sample points before passing them to the corresponding object NeRF.

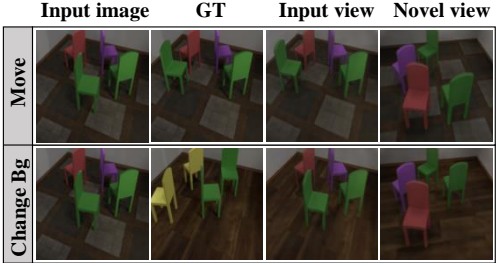

Figure 4: 3D scene manipulation for moving object and changing background.

Meanwhile, for changing the background, we substitute the original background texture by replacing the original background slot feature with that of the target image.

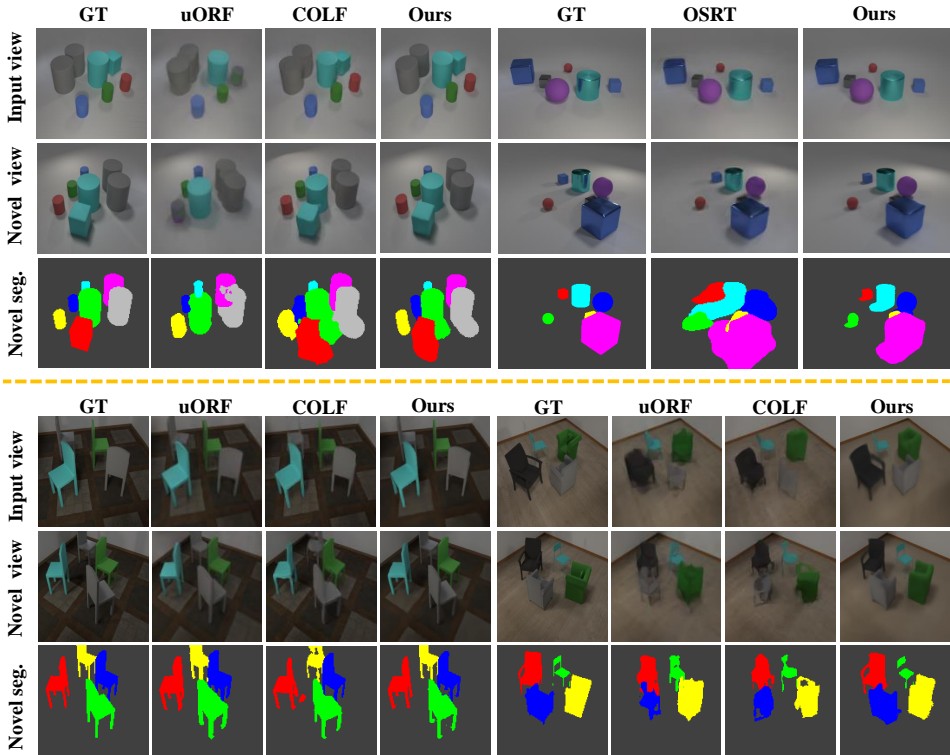

Figure 3: **Qualitative Comparison.** We compare the reconstructions of the input view, a novel view, as well as the novel view segmentation using the uORF [8], COLF [9], OSRT [3], and our method on four datasets. Our method produces a finer segmentation and more precise shapes.

**Results**  Figure 4 shows that edited images maintain a harmonious quality while accurately executing object movement and background replacement, affirming the strong correlation between slots and 3D objects in sVORF, as well as the accuracy of our model in 3D scene segmentation.

### 4.4 Ablation Studies

In this section, we conduct ablation studies on the CLEVR-567 dataset to gain a deeper understanding of how the various parts contribute to the overall effectiveness of our approach. More ablation studies on architectures are detailed in Appendix C.

Table 5: Ablation studies on the CLEVR-567 dataset.

| Model | NV-ARI ↑ | FG-ARI ↑ |
|---|---|---|
| sVORF (w/o NVS) | 15.1 | 31.2 |
| sVORF (w/o CR) | 81.1 | 89.3 |
| sVORF (density-weighted) | 81.4 | 88.6 |
| sVORF (w/o SA) | 79.6 | 85.4 |
| sVORF (ours) | **81.5** | **92.0** |

**Role of Novel View Synthesis**  To investigate the influence of the novel view synthesis setup on the training process, we modify our reconstructed target view to equal with the input view, *i.e.* , we turns sVORF into a 2D image auto-encoder. As shown in Figure 5, the model divides images based on areas rather than objects. This division led to substantially lower ARI results, presented in Table 5. Above observations confirm the crucial role of viewpoint changes in scene decomposition. Specifically, the mapping between slots and 3D representations captures changes occurring in various regions due to differences in viewing angles between the target and source views. Changes that occur within regions belonging to the same object are more closely approximated, which allows the model to converge to K slots based on object clustering.

**Connectivity Regularization**  As previously mentioned, novel view synthesis setup ensures distinctiveness between objects, while implementing Connectivity Regularization guarantees connectivity within the same objects. These two factors work together to achieve a clear decomposition of the scene.  Figure 5 visualizes that our Connectivity Regularization ensures that points on an object belong to the same slot by preventing semi-transparent clouds from being modeled by object NeRFs.

Numerically, as shown in Table 5, the model's FG-ARI displays a significant increase from 89.3 to 92.0 when implementing connectivity regularization compared to the model without it.

**Composing Mechanism**   As described in Section 3.3, our composing mechanism can perform well on small objects with few rays. To verify this, we implement the common density-weighted mean combination mode on our model and present the visualization in Figure 5. Our composing mechanism outperforms the density-weighted mean combination mode in decoupling effect, especially for small objects that may otherwise be segmented into attachments of other objects.

**Self-Attention in Scene Decomposition**   If an object in the input image is occluded, it may be mistakenly segmented into the slot of another object that corresponds to the occluded area. This issue can be resolved by leveraging the self-attention layer in the decomposition module as it facilitates the interaction between slots. As illustrated in the Figure 5, removing the self-attention layer leads to incorrect division of the occluded object, highlighting the importance of inter-slot interaction in preventing the collapse of the corresponding slot of the occluded object.

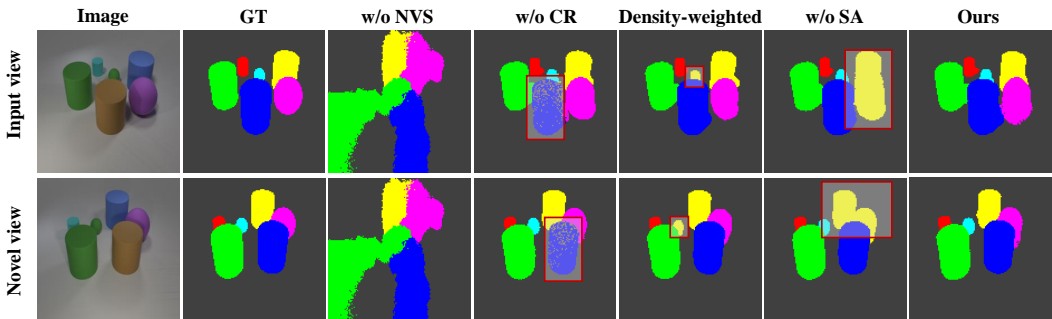

Figure 5: Qualitative comparison of ablation studies on CLEVR-567 dataset. Specifically, we evaluate the impact of four techniques - Novel View Synthesis (NVS), Connectivity Regularization (CR), Composing Mechanism, and Self-Attention (SA).

## 4.5   Generalization

We conduct three sets of generalization experiments. The three subsets assess the ability of sVORF to generalize to unseen object appearances, unfamiliar spatial arrangements and grayscale images respectively.

**Object Appearance**   For unseen object appearance, we follow the dataset in uORF, which is similar to CLEVR-567. The training set of this dataset excludes red cylinders and blue spheres, while the testing set includes only these types of objects. The results are presented in Table 6. It is noteworthy that our approach has never encountered the appearance of test objects but achieves comparable results to the models trained on a standard CLEVR-567 dataset.

Table 6: Object Appearance

| Model | ARI ↑ | Fg-ARI↑ |
| --- | --- | --- |
| Slot-Attention [43] | 2.2 | N/A |
| uORF [8] | **85.5** | N/A |
| sVORF | 82.0 | 92.6 |
| sVORF(567) | 83.9 | **95.8** |

**Spatial Arrangements**   To assess the model's ability to generalize to a greater number of objects with unseen, challenging arrangements, we train our method using 11 slots on the CLEVR-567 dataset and evaluate on the packed-CLEVR-11 dataset provided by uORF. During training, we randomly mask off four slots to prevent slot collapse, but we utilize all slots during testing. The

Table 7: Spatial Arrangements

| Model | ARI ↑ | Fg-ARI↑ |
| --- | --- | --- |
| Slot-Attention [43] | 5.7 | N/A |
| uORF [8] | **83.2** | N/A |
| sVORF | 81.0 | **85.5** |

performance of our method on the unseen object arrangements test set is shown in Table 7 and is found to be reasonably well.

**RGB color** To explore whether the sVORF mainly relies on RGB color for scene decomposition, we conduct an evaluation on a grayscale version of CLEVR-567 dataset. The model used in the evaluation is only trained on RGB CLEVR-567 dataset. The model achieves 87.5 FG-ARI on the grayscale test set, which is on par with 92.0 FG-ARI on the default RGB images. The evaluation results demonstrate that sVORF really learns to decompose the scene intrinsically. For qualitative results, see details in Appendix D.

### 4.6 Object Segmentation on Real Images

We demonstrate the effectiveness of our approach in real-world scenarios by validating it on LLFF datasets. To handle complex scenarios, we implement VIT-Base [47] as the backbone instead of ResNet34 [48] and train the network from scratch. Qualitative visualizations displayed in Figure 6 show that sVORF effectively segments the object within the scene, demonstrating the potential of our approach to understand general real scenarios.

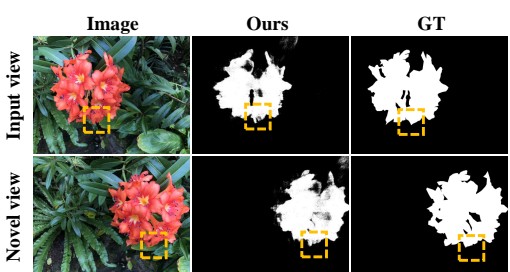

Figure 6: Object segmentation on real images.

### 4.7 Training Speed and Memory Consumption

**Setup** We compare sVORF with uORF [8] and COLF [9] in terms of memory consumption and training speed. For all three methods, we train models on the CLEVR-567 dataset using a batch size of 1 on V100 and record their memory usage and time taken for one epoch. Since both COLF and uORF render and supervise images at $64 \times 64$ resolution to learn the coarse structure first before supervising them at $128 \times 128$ resolution, we record their performance during both stages.

**Results** As shown in Table 8, sVORF offers substantially shorter training time (3m24s vs. 9m2s) and lower memory consumption (4.6G vs. 23.7G) than the volumetric-decoder based uORF method, while displaying comparable performance to the light field-based COLF.

Table 8: Memory Consumption and Performance Comparison.

| Model | Volumetric Field | | | Light Field | |
|---|---|---|---|---|---|
| | sVORF | uORF [8] | | COLF [9] | |
| | | coarse | fine | coarse | fine |
| Memory | 4.6G | 23.6G | 23.7G | 2.8G | 3.8G |
| Training Time | 3m24s | 8m42s | 9m2s | 1m56s | 2m22s |

In terms of total training time, for the CLEVR-567 and Room-Chair datasets, we train sVORF for approximately 7 hours using 8 Nvidia RTX 2080 Ti GPUs with batch size 16. The uORF and COLF models are trained on an Nvidia RTX V100 GPU for approximately 7 and 2 days, respectively, with a batch size of 1. For the CLEVR3D dataset, sVORF is trained for approximately 2 days using 8 Nvidia RTX V100 GPUs with batch size 16, while OSRT is trained for approximately 1 day on 8 A100 GPUs with a batch size of 256. These results demonstrate the effectiveness of sVORF in overcoming the challenges of volumetric decoders that demand extensive resources, while also contributing to enhanced training efficiency.

## 5 Conclusion

We present sVORF, a novel method for 3D object-centric representation learning. By adopting volumetric rendering to synthesis novel views and using object slots as a guidance to compose volumetric object radiance fields, sVORF precisely decompose 3D scenes into individual objects with limited training resources. It significantly outperforms existing SOTA methods, particularly in complex multi-object scenes. In addition, we demonstrate sVORF's ability for realistic scenario decomposition, indicating a promising direction for understanding the physical world.

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

# A Additional Training and Architecture Details

## A.1 Training details

We utilize the Adam optimizer with a learning rate of 0.0001, $\beta_1 = 0.9$, and $\beta_2 = 0.999$. Additionally, we implement learning rate warm-up for the initial 1,000 iterations. The minimum number K of objects in each scene is customized separately as follows: K = 8, 7, 5, 5, 2, and 5 for the CLEVR-567, CLEVR-3D, Room-Chair, Room-Diverse, LLFF, and MSN datasets, respectively.

To allow training on a high resolution, such as $256 \times 256$, we render individual pixels instead of large-sized patches. Specifically, we randomly sample a batch of 64 rays from the set of all pixels in the dataset, and then follow the hierarchical volume sampling [4] to query 64 samples from the coarse network and 128 samples from the fine network.

In addition, we train our model from scratch, with the exception of the Room-Diverse dataset. The Room-Diverse dataset is more complex and requires an incremental learning approach. Specifically, we initialize our models for Room-Diverse using weights from a model that have previously been trained on the CLEVR-567 dataset.

## A.2 Architecture details

**Image encoder** For the image encoder $E$, we utilize the well-known ResNet34 [48] architecture as the backbone, followed by three upsampling layers. Specifically, given the source image $\mathbf{I} \in \mathbb{R}^{256 \times 256 \times 3}$, we extract features $\mathbf{F}_1 \in \mathbb{R}^{128 \times 128 \times 64}$, $\mathbf{F}_2 \in \mathbb{R}^{64 \times 64 \times 128}$, and $\mathbf{F}_3 \in \mathbb{R}^{32 \times 32 \times 256}$ from the $conv2$, $conv3$ and $conv4$ layer of the ResNet34 architecture, respectively. Note that the max-pooling layer in $conv2$ is not used. Subsequently, these features are fed into the following three up-sampling layers within a U-net expansive path, resulting in the feature $\mathbf{F} \in \mathbb{R}^{128 \times 128 \times 512}$. The architecture of upsample layers is shown in Table 9.

Table 9: The architecture of upsample layers. All convolutional kernel sizes are $3 \times 3$. All activation functions for convolutional layers are ReLU. "+" indicates channel concatenation with a feature map of same resolution sourced from ResNet34.

| Layer name | Input shape | Output shape | Stride |
|---|---|---|---|
| Conv1 | $32 \times 32 \times 256$ | $32 \times 32 \times 512$ | 1 |
| Bilinear Upsampler | $32 \times 32 \times 512$ | $64 \times 64 \times 512$ | |
| Conv2 | $64 \times 64 \times (512 + 128)$ | $64 \times 64 \times 512$ | 1 |
| Bilinear Upsampler | $64 \times 64 \times 512$ | $128 \times 128 \times 512$ | |
| Conv3 | $128 \times 128 \times (512 + 64)$ | $128 \times 128 \times 512$ | 1 |

**Object NeRF** To represent each object NeRF, we employ a simple 4-layer MLP that each layer has 128 channels and is followed by a ReLU activation. In addition, we use skip connection mechanism that add the first layer's activation to the third layer's activation. Note that the last layer outputs the RGB value with a sigmoid activation function.

**Transformer-based module** The efficient transformer module is a standard transformer decoder. Specifically, we firstly use each slot $\mathbf{z}_i$ as query and interact with other object slots with multi-head self-attention operation. Then we employ multi-head cross-attention operation to attend into and aggregate features from the flattend image features $E(\mathbf{I})$. Finally, we pass the resulting slot features into a feed forward network (FFN) to get the final slots. This transformer module is simple and easy to train than the slot attention module, as it does not contain a Gated Recurrent Unit (GRU) block.

**Composition module** The aggregation block performs a cross-attention operation, which aggregates object representations $S$ with the 3D location $x$ as the query to obtain the corresponding feature $z$. The attention block computes the similarity between $S$ and $z$ after mapping them into the same space through a linear layer, thus obtaining the probability distribution of $x$ belonging to each slot. In the initial stages of our experiments, we observed that utilizing both modules simultaneously yielded superior results. We speculate that this improvement may be attributed to a good feature space alignment between 3D points and slot features using the proposed aggregation block.

## B  Object Segmentation on Real Images

**Setup**    In this study, we use VIT-Base [47] as the feature extractor for complex datasets of size 1008×756 with intricate textures. The adopted backbone networks are trained from scratch, and the method strictly follows unsupervised learning. To conserve memory, we resize the source image to 256× 256 and select only 7168 rays from the target view during each training iteration. Moreover, we define the task as foreground-background segmentation. As such, we set the number of slots to 2 and map it to two NeRFs. Each NeRF consists of a simple 4-layer MLP.

Unlike the setup on other datasets, we train sVORF on both LLFF scenes as two separate models Following the setup in NeRF-SOS, we divide the data of each scene into training and testing sets, ensuring there is no overlap between these two sets.

**Results**    Figure 7 shows the complete segmentation results of sVORF on both *Flower* and *Fortress* datasets. The results demonstrate the potential of sVORF for 3D segmentation on non-object-centric real scenes with cluttered backgrounds. Additionally, we use ResNet34 as the backbone and provide the segmentation results in Figure 8. Unlike sVORF with ViT-Base, sVORF with ResNet34 produces a coarse segmentation and still segments foreground object from complex scenes.

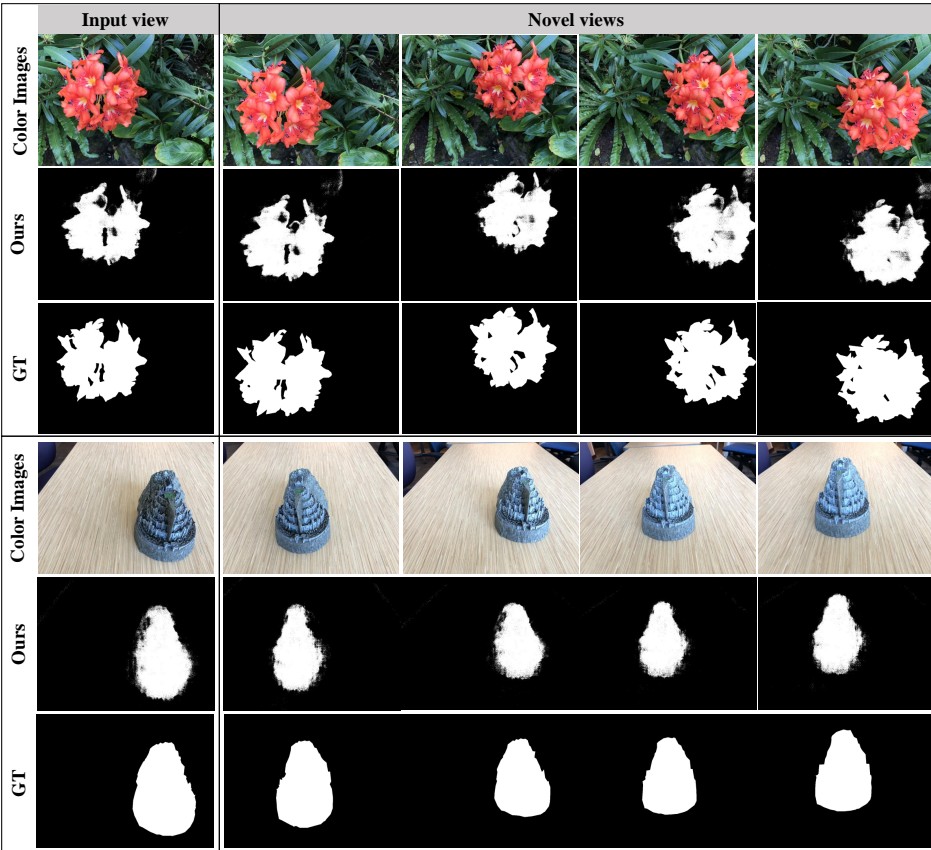

Figure 7: Qualitative results of 3D segmentation on real images.

## C  More Ablation Studies

This section provides more ablation studies on architecture, assessing the efficacy of the transformer-based module (Section 3.2), hypernetwork (Section 3.3) and combination module (Section 3.3), respectively. The quantitative results are reported in Table 10.

**Transformer-based module**    We substitute the transformer with slot attention and observe that slot attention fails to achieve the decomposition task in our model, as shown in the second row of Table 10.

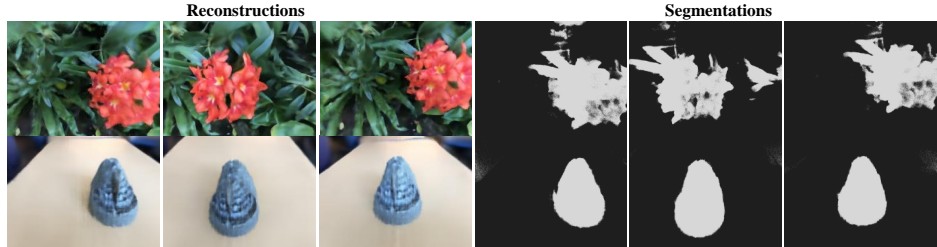

Figure 8: Qualitative results of sVORF with ResNet-34 on LLFF dataset.

Table 10: Ablation studies on the CLEVR-567 dataset.

| Model | NV-ARI ↑ | FG-ARI ↑ |
|---|---|---|
| sVORF (w/o Hypernetwork) | 21.6 | 65.9 |
| sVORF (w Slot-Attention) | 14.1 | 76.8 |
| sVORF (w SM) | 28.4 | 71.2 |
| sVORF (ours) | **81.5** | **92.0** |

Based on this comparison, we can conclude that our transformer-based module has a better scene decomposition than slot attention in our training setting.

**Hypernetwork**    We replace the hypernetwork with directly using the slots for conditioning the radiance fields per object like uORF [8]. As shown in the first row of Table 10, the model's performance significantly decreases. We speculate that using hypernetwork can provide stronger 3D geometric bias than directly using the slots for conditioning the radiance fields per object.

**Our composition module v.s. Slot Mixers**    To compare our composition method with Slot Mixers decoder, we have some modifications on the SM decoder [3]. First, we use a 3D point $x$ instead of the target ray as the query to aggregate the weighted slot feature. Second, we transform the weighted slot feature into the corresponding radiance field. Third, based on the radiance field, we can obtain the density and color of the 3D point $x$. In the experiment, we find that the composition performance of the SM method is lower than our proposed composition method. Specifically, the SM method exhibits 3D-inconsistent segmentation results, as shown in the third row of Table 10. It proves that the introduction of 3D geometric bias is really important for scene decomposition.

## D    More Qualitative Results

**Depth maps**    We illustrate the depth maps of sVORF on different datasets in Figure 9. The results show that our method can learn a high quality of 3D geometry.

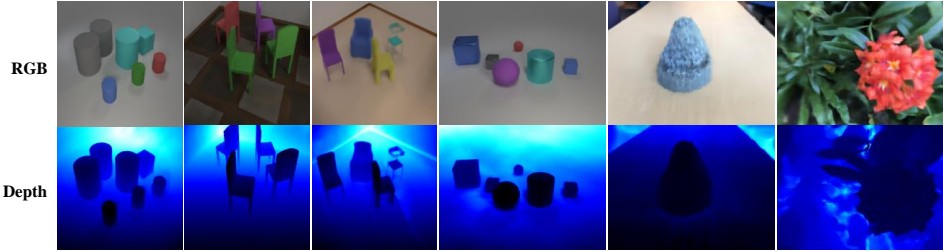

Figure 9: Reconstructed depth maps of sVORF on different datasets.

**Object-level radiance fields.**    We provide the visualization of learned object-level radiance fields on CLEVR-567 dataset in Figure 10. It is further demonstrated that our method can achieve very clean scene decomposition.

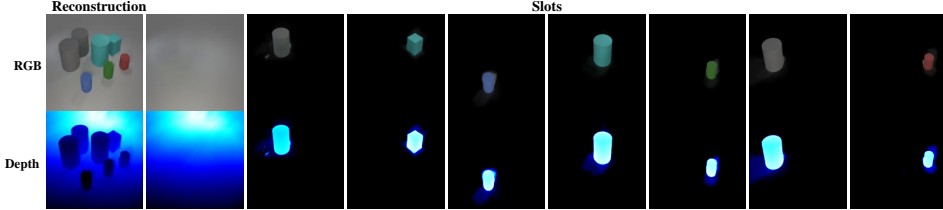

Figure 10: Visualization of learned object-level radiance fields.

**Grayscale images.** We provide the multi-view results of the qualitative evaluation of the model on a grayscale version of the CLEVR-567 dataset in Figure 11.

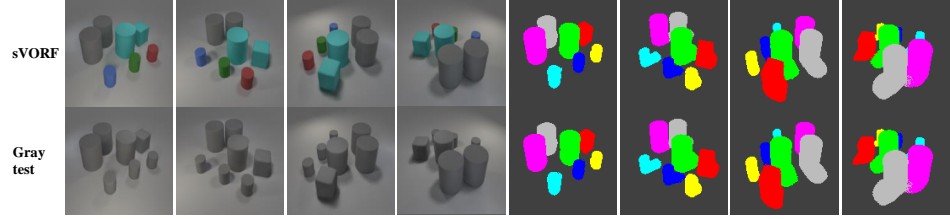

Figure 11: Multi-view qualitative results on a grayscale version of the CLEVR-567 dataset.

# E    Limitations

There are three limitations in sVORF. First, although sVORF can generalize to more objects at test time, as shown in Section 4.7, the maximum slot number of test scenes is restricted by the training setting. In other words, we need know the maximum number of objects/slots in the scene, and ensure that the number of slots set is equal to or exceeds this maximum value. Second, like other 3D representation learning methods, the model's training process requires curated multi-view images, which are difficult to obtain and necessitate specialized equipment, particularly in real scenes. Finally, although our method performs well on some complex scenes, such as MSN, it is still a challenge for our method to decompose and understand real-world scenes like a human. Addressing the limitations related to the slot number, training dataset and the generalization capabilities on real scenes is a potential area for future research.

