# OpenReview forum: "Slot-guided Volumetric Object Radiance Fields"
_NeurIPS.cc/2023/Conference — NeurIPS 2023 poster_

### Official Review · Reviewer_6zB4 · 2023-07-03

**Soundness:** 2 fair
**Presentation:** 3 good
**Contribution:** 2 fair
**Rating:** 4
**Confidence:** 2

**Summary:**

This paper presents slot-guided volumetric object radiance fields (sVORFa), a learning method for 3D object-centric representation. They proposed three key techniques, scene decomposition based on global pooling and slot-based representations, hyper-network to generate per-object radiance fields, and scene composition module. They have tested the method on various tasks on synthetic datasets, such as CLEVR and Room, showing superior performance to prior arts. Finally, they provided the simple experiment results on real images (LLFF), highlighting the broader effectiveness of the methods.

**Strengths:**

1. The proposed method improved the prior methods by a large margin, especially on Room-Chair and Room-Diverse datasets (segmentation task)
2. The newly suggested modules seem very efficient. and I think the global pooling features combined with the slot-guided method is a reasonable and better approach than the previous slot-attention method, at least in these particular tasks.


**Weaknesses:**

1. From my understanding, three modules were newly adopted or introduced to object-centric representation learning tasks. They are 1. ‘global pooling’ with slot-guided scene decomposition module, 2. hypernetwork, and 3. scene composition modules. However, throughout the experiments, I do not see which components are actually effective compared to prior methods. For example, they could have introduced ‘hypernetwork’ to uORF, or they could have tested ‘slot-attention’ in the proposed sVORF to see the effectiveness of the 'hypernetwork' or ‘global-pooling’ slot-guide method. Hence, it is hard for the readers to know every component actually contributes to the final performance.
2. It seems connectivity regularization (CR) is very important, I would like to see how much CR can improve the performance of the previous methods, such as uORF and others.
3. Similarly, I believe we can easily incorporate spatial broadcast (SB) or slot mixers (SM) in the proposed method sVORF, which could clearly show the effectiveness of the proposed scene composition module.
4. Although the newly introduced modules are well orchestrated into a single model and improve performance, I do not see significant technical novelty.

**Questions:**

1. In Table 2, why did you have to use different settings and re-evaluate the previous methods?

Other questions are embedded in the weaknesses section.

**Limitations:**

I do not see any potential negative societal impact from this work.

---

> ### Author Rebuttal · Authors · 2023-08-09
>
> Thank you for your careful review and helpful suggestions!
>
> $\textbf{Q1:}$ Need ablation studies on main components.
>
> $ \textbf{A1:}$ We conduct some experiments to compare our proposed components with previous methods. $\textbf{Firstly}$, we show the effectiveness of the transformer-based module. We substitute the transformer with slot attention and we observe that slot attention fails to achieve the decomposition task in our model, as shown in Fig 5 in the rebuttal PDF. Based on this comparison, we can conclude that our transformer-based module has a better scene decomposition than slot attention in our training setting.
> $\textbf{Secondly}$, we validate our proposed slot-guided scene composition module. We replace the hypernetwork with the conditional NeRF used in uORF. As shown in Table of our global response, using a hypernetwork performs better than using conditional NeRF. We speculate that using hypernetwork can provide stronger 3D geometric bias than directly using the slots for conditioning the radiance fields per object. Besides, we demonstrate the efficacy of using slots as a guidance to compose individual objects and background. As shown in section Composing Mechanism, this scheme outweighs the density-weighted mean used in uORF largely in FG-ARI metric. This comparison proves that slot-guided composition can make slot features 3D-aware, which is useful for scene decomposition.
>
> $\textbf{Q2:}$ How much CR can improve the performance of the previous methods, such as uORF and others.
>
> $\textbf{A2:}$ The CR module is not the key to sVORF decoupling. It serves the purpose of mitigating the presence of semi-transparent clouds when the number of slots significantly exceeds the total number of objects in a scene, as discussed in section 3.4 of main paper. We also give an analysis of how this module affects the overall performance of the model in section 4.4. Based on the analysis, it is not one of the key components. Considering that the occurrence of semi-transparent clouds is infrequent in uORF, it is not necessary to provide experimental results of CR module on uORF.
>
> $\textbf{Q3:}$ Incorporate Spatial Broadcast (SB) or Slot Mixers (SM) in the proposed method sVORF.
>
> $\textbf{A3:}$ For SB method, we present the ablation results in the Composing Mechanism section of the main paper. The results show that our composing mechanism outperforms the density-weighted mean combination mode (SB), especially for small objects that may otherwise be segmented into attachments of other objects.
> As for the SM method, we have some modifications on the SM decoder[1]. First, we use a 3D point $x$ instead of the target ray as the query to aggregate the weighted slot feature. Second, we transform the weighted slot feature into the corresponding radiance field. Third, based on the radiance field, we can obtain the density and color of the 3D point $x$. In the experiment, we find that the composition performance of the SM method is lower than our proposed composition method. Specifically, the SM method exhibits 3D-inconsistent segmentation results, as shown in Fig 5 in the rebuttal PDF. It proves that the introduction of 3D geometric bias is really important for scene decomposition.
>
> $ \textbf{Q4:}$ Limited significant technical novelty.
>
> $ \textbf{A4:}$ Unlike the previous work like uORF, our method has two unique contributions. $\textbf{Firstly}$, we use a transformer-based module instead of the GRU module to extract object and background slots. This module is simple and easy to optimize without GRU block. When we substitute the transformer with slot attention, we observe that slot attention fails to achieve the decomposition task in our model, as shown in Fig 5 in the rebuttal PDF. Based on this comparison, we can conclude that our transformer-based module has a better scene decomposition than slot attention in our training setting. $\textbf{Secondly}$, we propose a slot-guided scene composition module to recompose the slots to novel views. Compared to a conditional NeRF in uORF, this module uses a hypernetwork to transform a slot to its radiance field and utilizes explicit geometric bias to obtain density and color. This design performs better than using conditional NeRF, as shown in Table in our global response. We speculate that using hypernetwork can provide stronger 3D geometric bias than directly using the slots for conditioning the radiance fields per object. Besides, this module uses slots as a guidance to compose individual objects and background. This scheme can make slot features 3D-aware, which is useful for scene decomposition. As shown in section Composing Mechanism, this scheme outweighs the density-weighted mean used in uORF largely in FG-ARI metric.
>
> $\textbf{Q5:}$ In Table 2, why did you have to use different settings and re-evaluate the previous methods?
>
> $\textbf{A5:}$ In the CLEVR-3D dataset, ObsSuRF and OSRT employ distinct training and test data divisions, as well as different test metrics. To ensure a fair comparison, we use the settings of ObsSuRF as default and re-evaluated OSRT accordingly.
>
> [1] Mehdi SM Sajjadi, Daniel Duckworth, Aravindh Mahendran, Sjoerd van Steenkiste, Filip Pavetic, Mario Lucic, Leonidas J Guibas, Klaus Greff, and Thomas Kipf. Object scene representation transformer. Advances in Neural Information Processing Systems, 35:9512–9524, 2022.

---

### Official Review · Reviewer_thZt · 2023-07-04

**Soundness:** 3 good
**Presentation:** 4 excellent
**Contribution:** 3 good
**Rating:** 5
**Confidence:** 3

**Summary:**

This paper proposes a method, sVORF, for unsupervised object-centric 3D representation learning. Given a single image as input, it decomposes the scenes into individual 3D objects and 3D backgrounds which thus support object segmentation, novel view synthesis (named scene generation in the paper), and scene editing (including object moving and background repainting).
The key idea of the paper is to combine object slot with object radiance fields together in a framework, in which the object slot is responsible for decomposing the scene into objects. In contrast, the object radiance fields are used to re-compose the scene for volumetric rendering. The only cue for training the network is the rendering loss between the re-composed and volumetric rendered image with the input single-view image. The overall performance of sVORF is plausible and thorough experiments and ablation studies well validated the effectiveness of the proposed method.

**Strengths:**

+The task of reconstructing each object and the background in 3D given only a single RGB image is very challenging and with great importance for practical applications in robotics and CHI, etc.
+The overall performance of the proposed method is strong when compared with existing approaches.
+The idea of using the attention mechanism for scene decomposition and re-composition as well as the use of HyberNetwork for object-level NeRF generation is sound and practical.
+The demonstration of segmentation on real captured dataset LLFF is important to help the readers to qualify the performance of the method.
+The paper is well written and easy to follow.

**Weaknesses:**

- The dataset used for validating the proposed method is quite simple, with obvious color or shape differences between objects and simple backgrounds, which is very convenient for segmentation and decomposition. In the LLFF Dataset, the segmented foreground color is also very different from the background. So maybe more complex scenes should be used to demonstrate the effectiveness of the proposed method.
- The HyperNetwork produces MLP parameters of NeRF. However, the MLP representation of NeRF is very ambiguous, so learning such a hyberNetwork for NeRFs may not be very robust, please explain how to deal with this problem, especially for real-world scenes with complex objects in it.
- In Fig.5, only segmentation results are provided on the LLFF dataset, is it possible to also provide the NVS results? why not?
- The main paper should clarify the training strategies and key implementation details like how to set K for each dataset.

**Questions:**

- Line 224, please provide the full name of FFN since this is the first time you use FFN.
- In Table 2(a), it seems ObSURF works generally better on 3D segmentation, does this mainly benefit from the depth input?
- Line 149, a->an

**Limitations:**

Yes.

---

> ### Author Rebuttal · Authors · 2023-08-09
>
> Thank you for your careful review and helpful suggestions!
>
> $\textbf{Q1:}$ Demonstrate the effectiveness of the proposed method on more complex scenarios.
>
> $\textbf{A1:}$ To validate our method on more challenging scenarios, we conduct a preliminary experiment on the MSN dataset. The MSN dataset comprises $\textbf{11,733 distinct shapes}$, with each scene populated by 2-4 objects sourced from the ShapeNetV2 3D model dataset. The results are shown in below:
>
> | Model | Supervision |  ARI $\uparrow$ | Fg-ARI $\uparrow$ | PSNR $\uparrow$|
> | ----------- | ----------- | ------------ | ----------- | ----------- |
> | ObSuRF | image+depth  | $\bf{64.1}$ | 81.4 | 27.41 |
> | sVORF | image |  63.4 | $\bf{84.1}$ | $\bf{30.51}$ |
>
> Compared with ObSuRF, sVORF achieves significantly higher Fg-ARI and comparable ARI without using depth information, which demonstrates the model's ability to decouple more complex scenarios. Please refer to Fig 1 in the rebuttal PDF for visual results.
>
> $\textbf{Q2:}$ The robustness of hypernetwork.
>
> $\textbf{A2:}$ To our knowledge, Shap$\cdot$E[1] also learns a hypernetwork to produce MLP parameters. Shap$\cdot$E can handle a large complex and diverse 3D assets, although it does not involve complex background. It is an indicator that a hypernetwork for NeRF may be enough for real-world scenes with complex objects in it. For the success of Shap$\cdot$E, we speculate that a large dataset may be the key factor. Thus, to deal with the robustness of a hypernetwork, it may be an alternative to train with a large dataset. Certainly, we acknowledge the robustness problem of the hypernetwork. It is a future work to investigate and solve this problem.
>
> $\textbf{Q3:}$ Provide the NVS results on the LLFF dataset.
>
> $\textbf{A3:}$ As a result of limited page space, we provide the novel view synthesis results of LLFF in Appendix A.2. These results showcase the multi-view consistency of sVORF on the LLFF dataset. If possible, we will include these results in the main paper.
>
> $\textbf{Q4:}$ Clarify the training strategies and key implementation details like how to set K for each dataset.
>
> $\textbf{A4:}$ The implementation details are provided in Appendix. Furthermore, we employ additional training strategies as outlined below. We utilize the Adam optimizer with a learning rate of 0.0001, $\beta_1$ = 0.9, and $\beta_2$ = 0.999. Additionally, we implement learning rate warm-up for the initial 1,000 iterations. The value of K (set based on the number of objects in the scene) was customized separately as follows: K = 8, 7, 5, 5, 2, and 5 for the CLEVR-567, CLEVR-3D, Room-Chair, Room-Diverse, LLFF, and MSN datasets, respectively. For the CLEVR-567 and Room-Chair datasets, sVORF is trained with a batch size of 16  for approximately 7 hours on 8 Nvidia RTX 2080ti GPUs. For the larger CLEVR3D datasets, sVORF spends approximately 2 days using a batch size of 16 on 8 Nvidia RTX V100 GPUs.
>
> $\textbf{Q5:}$ Provide the full name of FFN.
>
> $\textbf{A5:}$ Thank you for catching that typo. The full name of FFN is feed-forward network.
>
> $ \textbf{Q6:}$ Influence of depth information on ObSuRF.
>
> $ \textbf{A6:}$ Regarding image generation, ObSuRF points out that incorporating additional depth supervision can significantly reduce reconstruction errors. As for decomposition, although ObSuRF does not provide ablation experiments about the depth map, it is obvious that depth information plays a crucial role in separating the foreground and background, particularly for shadows and foreground objects.
>
> $ \textbf{Q7:}$ Typo in L149
>
> $\textbf{A7:}$ Thank you for catching that typo and we will fix it in final version.
>
> [1] Jun, Heewoo, and Alex Nichol. "Shap-e: Generating conditional 3d implicit functions." arXiv preprint arXiv:2305.02463 (2023).

---

> > ### Comment · Reviewer_thZt · 2023-08-18
> >
> > Thanks for your reminder.
> > I am overall satisfied with the author's response. The effectiveness was evaluated using a more complex dataset. I would suggest discussing more insightfully the advantages of using hyper network but not simply following ShapeE. Moreover, the philosophy to set K should be clarified in the main paper to avoid confusion.

---

### Official Review · Reviewer_3Ksj · 2023-07-07

**Soundness:** 2 fair
**Presentation:** 3 good
**Contribution:** 3 good
**Rating:** 7
**Confidence:** 4

**Summary:**

This paper presents a method called sVORF for learning 3D object centric representations. A transformer decompose a single input image into slots which are then each mapped to volumentric radiance fields by a hypernetwork. These object-radiance fields are then composed into a 3D scene. The effectiveness of sVORF is demonstrated on a set of simple synthetic datasets and two real world scenes. It achieves SOTA or at least competitive results on CLEVR567, CLEVR3D, Room-Chair, and Room-diverse datasets, compared to SlotAttention, uORF and COLF.

**Strengths:**

* The problem of unsupervised 3d object decomposition is quite challenging. The fact that sVORF performs so well with only a single input image, and no depth or segmentation supervision is remarkable.
* The writing is clear and easy to follow. The figures look nice, are helpful and are easy to understand.
* The individual parts of the method are well motivated and their individual contributions are established through systematic ablations.

**Weaknesses:**

* The paper stresses the low computational costs and memory footprint of sVORF, but fails to report any data on them in the main paper. Memory consumption is mentioned in the appendix, alongside training time per epoch. But information about the length of training is lacking so the real computational costs remain unclear.

* The studied scenes are visually simple (yes even Room-diverse which is advertised as complex). The only exception are the two scenes from LLFF. It is encouraging that sVORF succeeds on these two scenes, but in this setting it appears to me that the model is effectively overfitting on the given scene, and there is no evidence for generalization to novel scenes in that setting.

* The restriction to a single input image is rather limiting. The 3D structure of a general scene is highly underspecified from a single image, which should lead to large uncertainties (e.g. in occluded areas). For simple scenes like CLEVR and Room-Chair this is not a problem because none of the elements is ever fully occluded, and they come from a small set of known objects. But I expect this to become a severe limitation for scaling to more complex scenes. Especially since sVORF and its training objective are not designed to be probabilistic / generative, and its fidelity will thus likely suffer substantially from complex ambiguities.

**Questions:**

- How many samples are used per scene during training. Appendix A mentions 64 rays with 64 coordinates in the coarse volume and 128 additional coordinates in the fine volume. What are the coarse and the fine volumes?
- For how long (number of updates / epochs) was sVORF trained? How does that compare to uORF, COLF, OSRT?
- The LLFF results use only two scenes. Does that mean the shown results are training images? Is sVORF trained on both scenes at once or as two separate models. In the latter setting it seems to me that sVORF is essentially equivalent to a simple volumetric NeRF, since the model could in principle learn to produce the correct target views during inference without even looking at the input view. Is there any evidence for generalization in this setting?

**Limitations:**

There is a brief discussion of limitations in the supplementary that touches on two important limitations of the method.
It does however not discuss any concerns regarding generalization to more complex real-world scenes.

---

> ### Author Rebuttal · Authors · 2023-08-09
>
> Thank you for your careful review and helpful suggestions!
>
> $\textbf{Q1:}$ Information about the length of training. How does that compare to uORF, COLF, OSRT?
>
> $\textbf{A1:}$ For the CLEVR-567 and Room-Chair datasets, we train sVORF for approximately 7 hours using 8 Nvidia RTX 2080 Ti GPUs with batch size 16. The uORF and COLF models are trained on an Nvidia RTX V100 GPU for approximately 7 and 2 days, respectively, with a batch size of 1.
> For the larger CLEVR3D datasets, sVORF is trained for approximately 2 days using 8 Nvidia RTX V100 GPUs with batch size 16, while OSRT is trained for approximately 1 day on 8 A100 GPUs with a batch size of 256.
>
> $\textbf{Q2:}$ The studied scenes are visually simple.
>
> $ \textbf{A2:}$ To validate our method on more challenging scenarios, we conduct a preliminary experiment on the MSN dataset. The MSN dataset comprises $\textbf{11,733 distinct shapes}$, with each scene populated by 2-4 objects sourced from the ShapeNetV2 3D model dataset. The results are shown in below:
>
> | Model | Supervision |  ARI $\uparrow$ | Fg-ARI $\uparrow$ | PSNR $\uparrow$|
> | ----------- | ----------- | ------------ | ----------- | ----------- |
> | ObSuRF | image+depth  | $\bf{64.1}$ | 81.4 | 27.41 |
> | sVORF | image |  63.4 | $\bf{84.1}$ | $\bf{30.51}$ |
>
> Compared with ObSuRF, sVORF achieves significantly higher Fg-ARI and comparable ARI without using depth information, which demonstrates the model's ability to decouple more complex scenarios. Please refer to Fig 1 in the rebuttal PDF for visual results.
>
> $\textbf{Q3:}$ The LLFF results use only two scenes. Does that mean the shown results are training images? On real-world datasets, there is no evidence that sVORF generalizes to new scenarios.
>
> $\textbf{A3:}$ We train sVORF on both LLFF scenes as two separate models, but the results presented do not include the training images. Following the setup in NeRF-SOS[1], we divide the data of each scene into training and testing sets, ensuring there is no overlap between these two sets.
> Our experiment on LLFF aims to demonstrate the capability of sVORF to segment objects in real scenarios. For a simple volumetric NeRF, it can not segment objects in this setting like sVORF. Due to the limited availability of multi-view data from real scenes, it is challenging to show generalization of our method in this setting. However, it is an interesting topic and we plan to investigate it in future research.
>
> $\textbf{Q4:}$ The restriction to a single input image is rather limiting.
>
> $ \textbf{A4:}$ We acknowledge that the restriction to a single input image is rather limiting, particularly when dealing with more complex scenes. Incorporating multi-view images as inference may alleviate this problem. We consider this as a potential avenue for future research.
>
> $ \textbf{Q5:}$ For how long (number of updates / epochs) was sVORF trained? How does that compare to uORF, COLF, OSRT?}
>
> $ \textbf{A5:}$ Please refer to the comments in A1.
>
> $ \textbf{Q6:}$ How many samples are used per scene during training?
>
> $ \textbf{A6:}$ We follow the hierarchical sampling strategy in NeRF[2]. Specifically, we sample 64 points per ray through the coarse network and 64 + 64 = 128 points per ray through the fine network. The "coarse" and "fine volume" refers to the “coarse” and “fine” network.
>
> [1] Zhiwen Fan, Peihao Wang, Yifan Jiang, Xinyu Gong, Dejia Xu, and Zhangyang Wang. Nerf-sos: Any-view self-supervised object segmentation on complex scenes. arXiv preprint arXiv:2209.08776, 2022.
> [2] Ben Mildenhall, Pratul P Srinivasan, Matthew Tancik, Jonathan T Barron, Ravi Ramamoorthi, and Ren Ng. Nerf: Representing scenes as neural radiance fields for view synthesis.  Communications of the ACM, 65(1):99–106, 2021.

---

> > ### Comment · Reviewer_3Ksj · 2023-08-18
> > **Response to rebuttal**
> >
> > First, I would like to thank the authors for writing such an extensive rebuttal with numerous additional experiments.
> > I agree with the other reviewers, in that the paper is more of a recombination and improvement upon previous works than an innovation in and of itself. I am also still skeptical about the ability of sVORF to generalize to data with real-world complexity.
> > However, I would like to point out that the problem of unsupervised scene decomposition is extremely challenging, and that the authors do demonstrate clear improvements above the baselines, especially in terms of compute. It is still not a strong result, but in my opinion the additional results and clarifications significantly improve the paper and push it into the region of a clear accept.

---

### Official Review · Reviewer_JyjT · 2023-07-08

**Soundness:** 3 good
**Presentation:** 3 good
**Contribution:** 2 fair
**Rating:** 6
**Confidence:** 5

**Summary:**

This paper introduces sVORF, a compositional method for representing scenes as a collection of slots that are parametrized as per-object radiance fields. The proposed model can be used to perform novel-view synthesis, as well as semantic segmentation in 3D. Moreover, the proposed compositional representation, also enables performing simple editing operations in the scene. In more detail, given an input image, sVORF first extracts image features from the image and passes them to a transformer encoder-decoder module that extracts slots for the objects in the scene and the background. Subsequently, these slots are mapped to volumetric object radiance fields using a hypernetwork. Using the per-object radiance fields, the scene is rendered by simply compositing their outputs at a 3D location with the guidance of each slot. Unlike [8] that was the first work that explored discovering slots in 3D data, sVORF maps slots to per-object radiance fields using a hypernetwork. The authors evaluate the performance of their model on the CLEVR-567, CLEVR-3D, Room-Chair, Room-Diverse and the LLFF datasets and compare with several compositional representations that represent slots as radiance fields [8, 35] or as light fields [9]. From their experimental results, it seems that the proposed model outperforms all baselines both on the novel view synthesis task as well as on the 3D segmentation task.

Overall, I think the proposed idea for discovering slots in 3D is interesting and the authors show that their approach is working for simple scenes with a few objects. In terms of novelty, the proposed pipeline is quite similar to [8], with the main difference being the use of the hypernetwork for mapping slots to per-object radiance fields. Looking at the quantitative results, it seems that the proposed model is better than the baselines on most tasks, thus I am leaning towards accepting the paper but I am still a bit concerned by the fact that the proposed model and [8] are relatively similar and that the proposed model is only evaluated on extremely simple scenes.

**Strengths:**

1. The authors conduct multiple experiments and compare their model with several baselines on multiple datasets. From the quantitative evaluation, we note that the proposed model outperforms all baselines on most tasks. I particularly liked the scene editing experiment in Sec. 4.3.

2. Although some things are not 100% clear I think the paper is easy to read and relatively nicely written. I really liked Figure 1 that is a pictorial representation of the overall pipeline.

**Weaknesses:**

1. One of the major weaknesses of this work is that the authors only demonstrate the performance of their model on very simple datasets that consist of few objects that in most cases belong to the same object class e.g. chairs. Although I am aware that all baselines evaluate their models on similar setups I think it would have been great if the authors could show that their model works on more challenging scenarios, with different objects and more diverse backgrounds. A rather simple indoor dataset that the authors could consider is the 3D-FRONT dataset. Another less exciting alternative would be to also show results on the City-Block dataset introduced in [9]. However, since also this dataset contains scenes with cars, I think that the 3D-FRONT would be a more appropriate benchmark.

2. In Sec. 3.2 and in L116-117 the authors state that the adopt an efficient transformer module to infer the object and the background slots form the image features. Can the authors describe how this efficient transformer module compares to the classical slot attention module used in [8]? Is the efficient transformer module a standard transformer encoder/decoder? I think that to make things very clear, it would have been very useful to properly define (e.g. using a math expression) how is the cross-attention defined? In addition, can the authors also clarify why/how is this model efficient?

3. Although, the authors provide several ablation studies, I believe that they should have ablated the impact of using a hypernetwork as opposed to directly using the slots for conditioning the per-object radiance fields. This is more similar to [8] and it could better explain why is the proposed model outperforming [8], although their quite similar.

4. I am bit concerned regarding the limited technical contribution of this work. Unless I am missing out something I think the proposed paper is quite similar to [8]. That being said, the proposed method seems to be significantly better than [8]. This could be either due to the use of the hypernetwork or the use of the attention module for inferring the slots. I believe that it is very important to clarify what are the differences and also try to justify the improvement in performance though ablations e.g. ablate the use of a hypernetwork for mapping slots to per-object radiance fields.

**Questions:**

1. Unless I am missing out something, I think that the proposed model assumes that the number of objects/slots is known right? In other words, although we might not know what are the objects, we know how many objects/slots exist in the scene, right? This is never clearly mentioned in the text. I think the authors should clarify it.

2. I think that the per-object NeRFs should be locally defined, namely each NeRF is augmented with a local affine transformation defining its pose in 3D. Otherwise, it would not be possible to change the location of a specific object/slot to a new position. Is this really the case?

3. I think that from the provided information for the composing mechanism in L146-150 it is not 100% clear how are the aggregation $D^c$ and the attention bloc $D^a$ really implemented. One thing that is not very clear to me is why do we need both? Can the authors please explain?

4. In L264-272, the authors present an ablation study that tries to investigate the impact of the Novel View Synthesis setup. To be honest, I am not 100% sure that I understand what the authors mean by "we modify our reconstructed target view to equal with the input view". Can the authors please clarify?

5. I am wondering what is the quality of the 3D geometry of the proposed model and how does it compare to baselines. Since the authors represent the scene using a compositional NeRF-based representation I think it would be useful to also show some depth maps at least in the supplementary.

**Limitations:**

The authors discuss the limitations of their work in their supplementary material but I was not able to find any discussion about the potential negative societal impact of their work. That being said, I think that for this paper this discussion is not 100% necessary.

---

> ### Author Rebuttal · Authors · 2023-08-09
>
> Thank you for your careful review and helpful suggestions!
>
> $\textbf{Q1:}$ Experiments on more challenging scenarios.
>
> $\textbf{A1:}$ To validate our method on more challenging scenarios, we conduct a preliminary experiment on the MSN dataset. The MSN dataset comprises $\textbf{11,733 distinct shapes}$, with each scene populated by 2-4 objects sourced from the ShapeNetV2 3D model dataset. The results are shown in below:
>
> | Model | Supervision |  ARI $\uparrow$ | Fg-ARI $\uparrow$ | PSNR $\uparrow$|
> | ----------- | ----------- | ------------ | ----------- | ----------- |
> | ObSuRF | image+depth  | $\bf{64.1}$ | 81.4 | 27.41 |
> | sVORF | image |  63.4 | $\bf{84.1}$ | $\bf{30.51}$ |
>
> Compared with ObSuRF, sVORF achieves significantly higher Fg-ARI and comparable ARI without using depth information, which demonstrates the model's ability to decouple more complex scenarios. Please refer to Fig 1 in the rebuttal PDF for visual results.
>
> $\textbf{Q2:}$ Details of the efficient transformer module.
>
> $\textbf{A2:}$ The efficient transformer module is a standard transformer decoder. Specifically, we firstly use each slot $\textbf{z}_i$ as query and interact with other object slots with multi-headed self-attention operation. Then we employ multi-headed cross-attention operation to attend into and aggregate features from the flattend image features $E(\textbf{I})$. Finally, we pass the resulting slot features into a feed forward network (FFN) to get the final slots.
> As mentioned in the main paper, this transformer module is simple and easy to train than the slot attention module, as it does not contain a Gated Recurrent Unit (GRU) block. Additionally, when we replace the transformer with slot attention, we observe that slot attention fails to achieve the decomposition task, see Fig 5 in the rebuttal. Based on this comparison, we can conclude that our transformer-based module has a better scene decomposition than slot attention in our training setting.
>
> $\textbf{Q3:}$ The impact of using a hypernetwork.
>
> $\textbf{A3:}$ To validate the effectiveness of the hypernetwork, we conduct an ablation study, which replaces the hypernetwork with directly using the slots for conditioning the radiance fields per object. As shown in Table in our global response and Fig 5 in the rebuttal PDF, the model’s performance significantly decreases. We speculate that using hypernetwork can provide stronger 3D geometric bias than directly using the slots for conditioning the radiance fields per object. This 3D-aware slot facilitates guiding the composition of all volumetric neural radiation fields.
>
> $\textbf{Q4:}$ The limited technical contribution
>
> Unlike the previous work like uORF, our method has two unique contributions. $\textbf{Firstly}$, we use a transformer-based module instead of the GRU module to extract object and background slots. This module is simple and easy to optimize without GRU block. When we substitute the transformer with slot attention, we observe that slot attention fails to achieve the decomposition task in our model, as shown in Fig 5 in the rebuttal PDF. Based on this comparison, we can conclude that our transformer-based module has a better scene decomposition than slot attention in our training setting. $\textbf{Secondly}$, we propose a slot-guided scene composition module to recompose the slots to novel views. Compared to a conditional NeRF in uORF, this module uses a hypernetwork to transform a slot to its radiance field and utilizes explicit geometric bias to obtain density and color. This design performs better than using conditional NeRF, as shown in Table in our global response. We speculate that using hypernetwork can provide stronger 3D geometric bias than directly using the slots for conditioning the radiance fields per object. Besides, this module uses slots as a guidance to compose individual objects and background. This scheme can make slot features 3D-aware, which is useful for scene decomposition. As shown in section Composing Mechanism, this scheme outweighs the density-weighted mean used in uORF largely in FG-ARI metric.
>
> $\textbf{Q5:}$ Is the number of slots known?
>
> $\textbf{A5:}$ We know the maximum number of objects/slots in the scene, and ensure that the number of slots set is equal to or exceeds this maximum value.
>
> $\textbf{Q6:}$ I think that each NeRF is augmented with a local affine transformation defining its pose in 3D. Otherwise, it would not be possible to change the location of a specific slot to a new position.
>
> $\textbf{A6:}$ Yes, as you mentioned, in order to relocate the slot, an affine transformation is applied to the 3D sample points before passing them to the corresponding object NeRF.
>
> $\textbf{Q7:}$ For composition mechanism, why both aggregation and attention modules are needed?
>
> $\textbf{A7:}$ The aggregation block performs a cross-attention operation, which aggregates object representations $S$ with the 3D location $x$ as the query to obtain the corresponding feature $z$. The attention block computes the similarity between $S$ and $z$ after mapping them into the same space through a linear layer, thus obtaining the probability distribution of $x$ belonging to each slot. In the initial stages of our experiments, we observed that utilizing both modules simultaneously yielded superior results. We speculate that this improvement could be attributed to the increased difficulty of directly computing the similarity between the 3D points and the slot feature when they are not spatially aligned using $D^a$.
>
> $\textbf{Q8:}$ Clarifing the meaning of "we modify our reconstructed target view to equal with the input view" in L264.
>
> $\textbf{A8:}$ It means that we turns sVORF into a 2D image auto-encoder.
>
> $\textbf{Q9:}$ The quality of the 3D geometry of the proposed model.
>
> $\textbf{A9:}$ As shown in the rebuttal PDF Figure 3, we illustrate the depth maps of sVORF on different datasets. The results show that our method can learn a high quality of the 3D geometry.

---

> > ### Comment · Reviewer_JyjT · 2023-08-21
> > **Rebuttal Acknowledgment**
> >
> > I would like to thank the reviewers for taking the time to provide additional results on more complex scenes as well as discuss the key differences between the proposed model and [8]. After having read the author's rebuttal and the other reviews, most of my concerns have been addressed, hence I will raise my score to 6: Weak Accept.
> >
> > That being said, I would like to urge the authors to include the additional experiments, provided in the rebuttal, in the final version of their paper. If possible, please also consider providing additional qualitative comparisons on the various datasets. Moreover, I think that the authors should also add a section discussing the differences of their model compared to [8]. I believe that the ablation study about the impact of the use of hypernetwork is very important and the authors should definitely include it in the final version of their paper.

---

### Official Review · Reviewer_uYMS · 2023-07-08

**Soundness:** 3 good
**Presentation:** 3 good
**Contribution:** 3 good
**Rating:** 5
**Confidence:** 3

**Summary:**

This paper proposes sVORF to tackle the challenging problem of ‘unsupervised’ scene decomposition by learning a series of slot-guided neural radiance fields. Inferring object-level NeRF from a hyper-network instead of costly GRU modules, the overall pipeline can be efficiently trained with on a collection of multi-view images and produce promising segmentation results on several synthetic benchmark datasets.

**Strengths:**


- The overall ideas is well motivated and easy to follow, and the main pipeline could be well taken.

- Extensive experiments on adequate synthetic datasets are conducted. The numerical results are promising  compared to SOTA baselines, with detailed ablations.

**Weaknesses:**

- Limited unique contribution compared to previous work like uORF.

   I think this paper proposes a good extension on top of uORF while my main concern is that the unique contributions are somewhat limited. The overall pipeline is built on top of previous work uORF by replacing the expensive GRU modules to hyper-network inferred compositional object-level NeRFs, which I see as the main difference bringing efficiency advantages.

- I do appreciate the promising performance on synthetic datasets, however I think more discussions and results towards the main statements (consistency, efficiency against current baselines) need to be further strengthened.
    - The method of using radiance fields instead of light-fields stand out in terms of strict multi-view consistency. Therefore, discussions and analysis of multi-view consistency of segmentation masks (on synthetic and real-world cases) are expected to show to validate this point.


- Are the adopted backbone networks (ResNet34 and ViT-Base of real-world image segmentation) pre-trained in any way or trained from scratch (L318)? As I am not sure if the used datasets are sufficient to train such large network. If so, is the method strictly unsupervised anymore? More clarifications are expected. Also, for real-world segmentation, I am wondering how does the backbone impact the performance if ResNet34 is used.

- How does appearance affect the perform? Does the mode really learn to decompose the scene intrinsically or a very good ‘colour/apperance’ segmentor? I see some good evidence from fig2 but find the networks struggles to separate close-by objects with similar colour as well as shadows. Moreover, related to last point of consistency, is this ‘incorrect pattern’ also consistent across view?


 - Related to last point, visualization of learned object-level radiance fields are expected. Readers could know if the method indeed decompose the scenes into objects and background as we expect.

- It would be more exciting to see how the methods work on real-world images with more than 1 foreground objects.

- What is the total training time required to converge sVORF as only per-epoch time is provided.

-L118. (GEU) --> (GRU)?

**Questions:**

Please see the weaknesses above.

**Limitations:**

Limitations have been discussed in the supplement.

---

> ### Author Rebuttal · Authors · 2023-08-09
>
> Thank you for your careful review and helpful suggestions!
>
> $\bf{Q1}$: Limited unique contribution compared to previous work like uORF.
>
> $\bf{A1}$:Unlike the previous work like uORF, our method has two unique contributions. $\textbf{Firstly}$, we use a transformer-based module instead of the GRU module to extract object and background slots. This module is simple and easy to optimize without GRU block. When we substitute the transformer with slot attention, we observe that slot attention fails to achieve the decomposition task in our model, as shown in Fig 5 in the rebuttal PDF. Based on this comparison, we can conclude that our transformer-based module has a better scene decomposition than slot attention in our training setting. $\textbf{Secondly}$, we propose a slot-guided scene composition module to recompose the slots to novel views. Compared to a conditional NeRF in uORF, this module uses a hypernetwork to transform a slot to its radiance field and utilizes explicit geometric bias to obtain density and color. This design performs better than using conditional NeRF, as shown in Table in our global response. We speculate that using hypernetwork can provide stronger 3D geometric bias than directly using the slots for conditioning the radiance fields per object. Besides, this module uses slots as a guidance to compose individual objects and background. This scheme can make slot features 3D-aware, which is useful for scene decomposition. As shown in section Composing Mechanism, this scheme outweighs the density-weighted mean used in uORF largely in FG-ARI metric.
>
> $\bf{Q2}$: Limited discussions and results towards the main statements (consistency, efficiency against current baselines).
>
> $\bf{A2}$: $\textbf{Multi-view consistency}$: To our knowledge, the generation and segmentation quality of novel view can show the multi-view consistency of the proposed method.  As shown in Table 1, our method beats all baselines in terms of NV-ARI (ARI on synthesized novel views) in both Room-Chair and Room-Diverse scene. Besides, our method outperforms other baselines on novel view synthesis, as shown in Table 3.
> $\textbf{Efficency}$: We provide the details about the training speed and memory consumption in Appendix A.1.
> In addition, we provide some examples of depth maps for each dataset to show the effectiveness of our method in Fig 3 of the rebuttal PDF.
>
> $\bf{Q3}$: Is the method strictly unsupervised anymore? How does the backbone impact the performance if ResNet34 is used in real-world cases?
>
> $\bf{A3}$: The adopted backbone networks are trained from scratch, and the method strictly follows unsupervised learning.
> We use ResNet34 as the backbone and provide the segmentation results in Fig 2 of the rebuttal PDF. Unlike sVORF with ViT-Base, sVORF with ResNet34 produces a coarse segmentation and still segments foreground object from complex scenes.
>
> $\textbf{Q4}$: How does appearance affect the perform? Is the ‘incorrect pattern’ of shadow also consistent across view?
>
> $\textbf{A4}$: To explore whether the sVORF mainly relies on RGB color for scene decomposition, we conduct an evaluation on a grayscale version of CLEVR-567 dataset. The model used in the evaluation is only trained on RGB CLEVR567 dataset. The model achieves 87.5 FG-ARI on the grayscale test set, which is on par with 92.0 FG-ARI on the default RGB images. The evaluation results demonstrate that sVORF really learns to decompose the scene intrinsically. For qualitative results , please refer to the rebuttal PDF.
> The ‘incorrect pattern’ of the shadow remains consistent across different views, as illustrated in Fig 5 in the rebuttal PDF.
>
> $\textbf{Q5}$: Visualization of learned object-level radiance fields.
>
> $\textbf{A5}$: Thank you for this suggestion! We provide the visualization of learned object-level radiance fields on CLEVR-567 dataset in Fig 4 in the rebuttal PDF. It is further demonstrated that our method can achieve very clean scene decomposition.
>
> $\textbf{Q6}$: Experiments on real-world images with more than 1 foreground objects.
>
> $\textbf{A6}$: To validate our method on more challenging scenarios, we conduct a preliminary experiment on the MSN dataset. The MSN dataset comprises $\textbf{11,733 distinct shapes}$, with each scene populated by 2-4 objects sourced from the ShapeNetV2 3D model dataset. The results are shown in below:
>
> | Model | Supervision |  ARI $\uparrow$ | Fg-ARI $\uparrow$ | PSNR $\uparrow$|
> | ----------- | ----------- | ------------ | ----------- | ----------- |
> | ObSuRF | image+depth  | $\bf{64.1}$ | 81.4 | 27.41 |
> | sVORF | image |  63.4 | $\bf{84.1}$ | $\bf{30.51}$ |
>
> Compared with ObSuRF, sVORF achieves significantly higher Fg-ARI and comparable ARI without using depth information, which demonstrates the model's ability to decouple more complex scenarios. Please refer to Fig 1 in the rebuttal PDF for visual results.
>
> $\textbf{Q7}$: The total training time.
>
> $\textbf{A7}$: For the CLEVR-567 and Room-Chair datasets, sVORF is trained with a batch size of 16 for approximately 7 hours on 8 Nvidia RTX 2080ti GPUs. For the larger CLEVR3D datasets, sVORF spends approximately 2 days using a batch size of 16 on 8 Nvidia RTX V100 GPUs.
>
> $\textbf{Q8}$: Typo in L118
>
> $ \textbf{A8}$: Thank you for catching that typo and we will fix it in the final version.

---

> > ### Comment · Reviewer_uYMS · 2023-08-18
> > **Response to author rebuttal**
> >
> > Firstly I thank authors for prodiving more clarifications and experiments on relatively more complext MSN dataset.
> > I have carefully read all the reviews and author response, overall I think some concerns have been succussfully addressed.
> >
> > I think the additional visualisation and analysis would be critical for people to see how the proposed methods decompose the scenes via slots related neural fields. In addition, I expect the detailed configuration as well as experiments on gray-scale images and MSN dataset to be fitted in the main paper and believe it will be a stronger submission.
> >
> >
> > However, some key limitations still remian, for example, I guess the proposed method as well as previous ones would struggle on real-world scenes with multiple objects with rich textures. I hope such limitations could be clearly discussed in the main paper to make it clear where the method stand in the challenging task of unsupervised scene decomposition.
> >
> > Overall, I raise my score to 5 with borderline acceptance considering above modifications and limitations.

---

### Author Rebuttal · Authors · 2023-08-09

We thank the reviewers for their helpful comments, according to the feedback and suggestions, we mainly added the following experiments:
- [uYMS, JyjT, 3Ksj, thZt] We conduct experiments on more challenging  MultiShapenet (MSN) dataset. The results demonstrate our model’s ability to decouple more complex scenarios.
- [uYMS, JyjT, 6zB4] We replace the hypernetwork with directly using the slots for conditioning the radiance fields per object, and observe that using hypernetwork can provide stronger 3D geometric bias than conditioning NeRF.
- [uYMS] We train sVORF with ResNet34 backbone on LLFF dataset, which produces a coarse segmentation and still segments foreground object from complex scenes.
- [JyjT, 6zB4] We replace the transformer with slot attention, and observe that our transformer decomposes scene better than slot attention.
- [6zB4] We incorporate Slot Mixers (SM) in our sVORF. The results prove that the introduction of 3D geometric bias in our slot-guided composition method is really important for scene decomposition.
- [uYMS] We discuss testing results on unseen grayscale version of CLEVR-567 dataset, indicating that our model really learns to decompose the scene intrinsically.
- [uYMS, JyjT] We provide some examples of depth maps for each dataset to show the geometry of our method.
- [uYMS] We give the visualization of learned object-level radiance field.

The results of the ablation experiments are recorded in the table below

| Model | NV-ARI$\\uparrow$  | FG-ARI $\\uparrow$ |
| --- | ----------- | ----------- |
| sVORF~(w/o Hypernetwork) | 21.6 | 65.9 |
| sVORF~(w Slot-Attention) | 14.1 | 76.8 |
| sVORF~(w SM) | 28.4 | 71.2 |
| sVORF~(ours) | $\bf{81.5}$ | $\bf{92.0}$ |

---

### Decision · Program_Chairs · 2023-09-21

**Decision:**

Accept (poster)

**Comment:**

This paper introduces Slot-guided Volumetric Object Radiance Fields (sVORF), a method that combines object-centric neural radiance fields with an attention-based encoder and hypernetworks to decompose an input image into individual object representations and render novel views of the underlying scene. The authors demonstrate that sVORF achieves competitive performance compared to established baselines in this area such as uORF or OSRT.

The paper is overall of high quality and addresses a challenging and interesting problem (3D scene decomposition) which is of relevance to the NeurIPS community. The reviewers appreciated that doing well on joint novel-view synthesis and (unsupervised) scene decomposition is very challenging, especially when doing so from only a single input view without using depth as auxiliary signal. sVORF shows clear advantages over prior work both in terms of decomposition / novel view synthesis (while not requiring depth or multi-view signals) as well as in terms of inference efficiency.

A primary concern by some of the reviewers is related to the technical novelty of the method: sVORF can be viewed as a combination of existing techniques / established principles (hypernetworks, transformers, and object-centric radiance fields), which might appear as not sufficiently novel for a conference like NeurIPS. In my view, the novelty of this work goes beyond the technical novelty of the individual techniques used in the method: as highlighted by the authors in the rebuttal, the finding that a standard transformer encoder suffices to learn object-decomposed representations in this setting without supervision and without the use of a specialized module like Slot Attention is remarkable and surely of interest for the community. The use of hypernetworks in this setting is also novel -- the authors clearly demonstrate that this architectural choice results in several quantitative benefits compared to prior work. Finally, as highlighted by reviewer 3Ksj, the problem of unsupervised 3D scene decomposition is very challenging (while being a problem of significant interest for the NeurIPS community), and progress / new findings in this area should be judged accordingly.

Altogether, I believe that this paper will be a valuable addition to the NeurIPS conference program, even though the method itself largely relies on a combination of established architectural components. I would like to ask the authors to include the modifications recommended by the reviewers when preparing the camera-ready version of the paper.